# Postnatal Development of the Circadian Rhythmicity of Human Pineal Melatonin Synthesis and Secretion (Systematic Review)

**DOI:** 10.3390/children11101197

**Published:** 2024-09-29

**Authors:** Ekkehart Paditz

**Affiliations:** Center for Applied Prevention®, Blasewitzer Str. 41, D-01307 Dresden, Germany; praxis@paditz.eu

**Keywords:** melatonin in infants, pineal gland, noradrenaline, pinealocytes, Lhx4-Homebox, chrononutrition, circadian rhythm

## Abstract

**Introduction:** According to current knowledge, at birth, the pineal gland and melatonin receptors are already present and the suprachiasmatic nucleus is largely functional, and noradrenaline, the key pineal transmitter, can be detected in the early foetal period. It is still unclear why the pineal gland is not able to start its own pulsatile synthesis and secretion of melatonin in the first months of life, and as a result, infants during this time are dependent on an external supply of melatonin. **Method:** The causes and consequences of this physiological melatonin deficiency in human infancy are examined in a systematic review of the literature, in which 40 of 115 initially selected publications were evaluated in detail. The references of these studies were checked for relevant studies on this topic. References from previous reviews by the author were taken into account. **Results:** The development and differentiation of the pineal gland, the pinealocytes, as the site of melatonin synthesis, and the development and synaptic coupling of the associated predominantly noradrenergic neural pathways and vessels and the associated Lhx4 homebox only occurs during the first year of life. **Discussion:** The resulting physiological melatonin deficiency is associated with sleep disorders, infant colic, and increased crying in babies. Intervention studies indicate that this deficiency should be compensated for through breastfeeding, the administration of nonpooled donor milk, or through industrially produced chrononutrition made from nonpooled cow’s milk with melatonin-poor day milk and melatonin-rich night milk.

## 1. Introduction

The pulsatile circadian synthesis and secretion of melatonin in the darkness of the night is an essential unique feature of the pineal gland within circadian timing functions [1,2]. The anatomical structure of the pineal gland is already present at birth. This chronobiological timer function of the pineal gland is realised in close interaction with the leading master clock, the suprachiasmatic nucleus (SCN) (see below in the text of the present work). The present study focuses on this pineal-chronobiological-circadian role of melatonin, which has developed in vertebrates and mammals in the evolution of life. Other locations of melatonin production, such the intestinal tract, the skin, and the retina, and the evolutionary function of melatonin as a radical scavenger, have been described in detail by others [3,4,5,6,7,8,9,10,11].

Noradrenaline is the leading transmitter that, by activating alpha1-, alpha2-, and beta1-adrenergic receptors of the pinealocyte membrane via cAMP and cGMP pathways, contributes to the activation of the pineal enzyme group, including timezyme (AANAT, Arylalkylamine N-acetyltransferase), and thus to the start of pineal melatonin synthesis [12] (p. 4 and Figure 5 in the cited paper), ref. [13] (Figure 3 in the cited paper); for detailed reviews, see [13,14,15]. Stimulation of beta1-adrenergic receptors leads to the activation of AANAT mRNA in the cell nucleus via cAMP, inducing the pineal synthesis of melatonin [13] (Figure 3 in the citated paper). Timezyme (AANAT) is the rate-limiting enzyme in melatonin synthesis. The fibres of the nervus conarii encase the pineal gland in a tight network of fibres. At night, noradrenaline is released to the pinealocytes to stimulate the synthesis of melatonin [13] (Figure 4 in the citated paper).

Although noradrenaline and melatonin receptors are detectable in the early foetal period (noradrenaline from the 5th week in the brain stem (pons/locus coeruleus) [16], and melatonin receptors in the foetal suprachiasmatic nucleus (SCN) from the 18th week [17]), infants are dependent on an external supply of melatonin via breast milk in the first months of life, as they are not yet able to synthesise melatonin despite the presence of the above-mentioned structural prerequisites [18].

This is not just a “physiological” maturation process without clinically tangible consequences, because non-breastfed infants show signs of clinical-anamnestic, circadian, biochemical, and moleculargenetic detectable chronodisruption [19], with an increased incidence of sleep disorders, infant colic, and increased crying [20,21]. The latter is associated with the dreaded shaken baby syndrome (non-accidental head injury) [22].

Stable circadian sleep–wake rhythms and increased synthesis and secretion of melatonin usually develop after the 2nd to 6th (or 17th) month of life:-In 1982, Hartmann et al. demonstrated **reduced postnatal plasma melatonin concentrations** in 26 male infants [23]. In 1987, Attanasio et al. showed that stable day–night rhythms with high nocturnal melatonin concentrations are only detectable from the age of 6 months [24]. In 1988, Waldhauser et al. reported reduced nocturnal melatonin concentrations in the first six months of life [25]. In 1996, Commentz et al. showed that 2 to 7-day-old infants with a gestational age of 26 to 42 weeks had only minimal melatonin concentrations without circadian rhythms [26]. In 2015, it was similarly pointed out that stable circadian rhythms, in terms of cortisol, melatonin, sleep, temperature, and the activity of circadian genes, only develop in the first 6 to 18 weeks after birth [27].-A circadian rhythm of the **excretion of sulphatoxy-melatonin in the urine** was not detectable in 26 male infants before the postnatal age of 12 weeks [23]. Kennaway et al. showed in healthy full-term infants that the excretion of this melatonin metabolite after midnight increased by five- to six-fold at the age of 9–12 months compared to the age of 6 weeks (08 +/− 103 vs. 2973 +/− 438 pmol/24 h) [28]. In preterm infants, this effect occurred 2–3 weeks later [21]. Children born in February or March showed significantly higher urinary sulphatoxy melatonin concentrations at night at the age of 8 weeks compared to children born in August or September. This seasonal effect is due to the prolonged period of darkness in the winter season, which results in a consecutive increase in melatonin synthesis and secretion. At the age of 16 weeks, these differences had levelled out [29]. Higher secretion rates in the urine during the evening hours (6:00 p.m. to 10:00 a.m.) were associated with an earlier onset of night-time sleep (r = 0.51; *p* < 0.05) [30].-Kate McGraw et al. combined the **diary method with temperature measurements** once a day for the first 6 months of life and, from the third week, with **saliva melatonin concentration measurements** once a week for 24 h. The child was breastfed as needed. Light exposure was controlled exclusively by natural sunlight. The child’s body temperature showed a circadian rhythm in the first few days of life, which stabilised after the first week. A circadian sleep–wake rhythm only developed between the 45th and 56th day of life (second month), which was associated with an increased melatonin concentration after sunset [31].-Kleitman and Hartmann recorded a free-running non-24-hour sleep–wake rhythm in 19 infants (10 boys, 9 girls) up to the age of 4 months using **actigraphy** [32]. Comprehensive current actigraphic measurements showed that 414 light- and dark-skinned infants (1.2% female, 65.2% black) from parents of different income levels exhibited a largely common trend, with more stable and longer nocturnal phases of motor rest developing only between the ages of 3 and 6 months [33]. Wulff et al. examined 12 infants using actigraphy at three points in time (1st–3rd, 7th–9th and 13th–15th weeks of life). A stable circadian rhythm was only detectable in the whole group at the age of 13–15 weeks [34].-Comparable empirical data had already been recorded by William Preyer (1841–1897) in his seminal work on **developmental psychology** in 1892 [35]. His son did not start sleeping through the night until the age of 17 months [36] (p. 106). In comparison to the “watered cow’s milk” and “sparse wet nurse’s milk” available at the time, feeding with breast milk was associated with longer sleep [36] (p. 105). In the fourth month, “persistent crying without a detectable cause” was still observed [36] (p. 420). His empirical physiological and psychological data, with precise circadian and age-related markers, are compatible with the above-mentioned results, in which melatonin measurements were included, and correspond to the current state of knowledge. He established empirical developmental psychology using the diary method [35].

Significantly higher melatonin concentrations can be detected in breast milk at night [18,20,22,37,38,39], which helps breastfed babies to sleep better, cry less, have less infant colic, and as a result, these babies are probably also less likely to be affected by shaking trauma.

**Intervention studies** involving the administration of tryptophan to infants aged 4–20 weeks have shown that they sleep better [40,41]. These trials suggest that tryptophan, as a precursor for the synthesis of serotonin and melatonin, may contribute to improve sleep in infants. Studies in which tryptophan was administered to infants in combination with melatonin and vitamin B6 for sedation prior to MRI or brainstem audiometry examinations [42,43] are not criticised in detail here, as these were not placebo-controlled studies, and the effects of the individual substances were not analysed in separate study arms.

Since these studies did not analyse the relationship between age and the effectiveness of the intervention, it is not possible to assess whether the administration of tryptophan as a precursor for melatonin synthesis is also effective in the first few weeks of life. However, the following critical considerations should be taken into account:1.The tryptophan scandal of the 1990s serves as a serious warning against the supplementation of industrial infant formula with tryptophan. A Japanese company had launched a tryptophan product on the market that was designed to help adults build stronger muscles. Numerous people fell ill with eosinophilia-myalgia syndrome and there were also deaths. Some data indicated that this preparation did not distinguish between L-tryptophan, the physiologically active substance, and differently configured racemates. The approval requirements were subsequently tightened internationally, so that a purity level of at least 97% L-tryptophan is required in tryptophan products [44,45,46,47,48]. In addition, tolerable upper limits for tryptophan supplementation were proposed for adults [49]. Subsequently, a case report was published on this clinical picture following significant overdoses of L-tryptophan [47].2.In addition, L-tryptophan can only cross the blood-brain barrier (BBB) if there is a defined concentration ratio between L-tryptophan and large neutral amino acids (LNAA) competing for BBB passage (LNAA = valine, leucine, isoleucine, histidine, lysine, methionine, threonine, tryptophan, and tyrosine) [22,50,51,52,53], and if sufficient albumin is available in the blood as a transport protein for L-tryptophan [52,54].3.Deaths in infants and young children have been documented in chronological association with significant overdoses without medical prescription [55,56,57], which indicates:
-that the melatonin concentrations in breast milk during the day and at night are likely to represent the gold standard for timing functions in infants, established over millions of years of evolution [22];-that melatonin should only be administered to infants under medical supervision, for justifiable indications [18,57,58,59,60], and in the lowest possible dosages [6,60,61]; and-that further basic research is needed to clarify whether, in addition to the greatly prolonged melatonin elimination half-life during infancy [18,62,63] (see discussion), other melatonin degradation pathways that have not yet been investigated [57,64] could be of significance in infancy.

In this context, the use of chrononutrition [22,41,65,66,67,68,69,70,71,72,73,74,75] with non-pooled day-night milk without external additions of L-tryptophan or melatonin [22,57,73] offers significant advantages. Non-pooled means that melatonin-poor day milk and melatonin-rich night milk are collected and administered separately. This applies to the administration of breast milk during breastfeeding, the administration of donor milk to premature babies via breast milk collection centres [75], and the production of industrial infant formula from non-pooled cow’s milk. The transition to such chrononutrition for infants represents a “back to nature” approach, ensuring that infants receive an adequate supply of melatonin in the first months of life [18,22,73].

This paper presents the current biochemical and pathophysiological knowledge on the postnatal development of pineal melatonin synthesis, the associated signal transduction chain from the retina via the suprachiasmatic nucleus (SCN) as the master clock for diurnal timing functions, and via the cervical ganglion to the provision of noradrenaline in the area of the pineal gland for the activation of pineal melatonin synthesis. The usefulness of chrononutrition with non-pooled milk is derived and justified from these basic pathophysiological facts and the above-mentioned intervention effects. It is examined as to whether the chronobiological hypotheses mentioned above can be substantiated from a pathophysiological point of view. The present study focuses on the reasons for physiological melatonin deficiency in the first months of life in infants in connection with the postnatal development of the pineal gland. On this pathophysiological basis, the clinical need for chrononutrition for infants is derived and justified.

## 2. Method

This systematic review was developed in accordance with the PRISMA checklist [76,77]: (1) Information on the author can be found above. (2) Not applicable, as no registration has been made. (3) The author confirms that the idea and the preparation of this review were carried out by him using the methodology described here. (4) There are no previous systematic reviews; however, the author has published current reviews on relevant topics [18,22,57,60,78,79]. (5) No financial support has been provided. (6) The topic of the present systematic review is derived and justified in the introduction presented here. (7) The questions to be examined in relation to this topic are based on clinical studies and reviews of clinical studies in humans. Data from animal experiments are used in the discussion if this allows important facts to be explored in greater depth. The limitations of transferring data from animal experiments to humans are explicitly pointed out. (8) All languages are included in the search. (9) Additional sources: see point 4. (10) The search strategy is described in the methodology section. (11a) The data from the search on 22 August 2024 were recorded via PubMed and sent to the author by email. The author saved the lists with the search results. (11b) and (11c) The procedure for the stepwise selection of the studies used is described in the method section. (12) The variables that were searched for are given in the method section. (13) With regard to the outcome variables, the search focuses on all available data that can contribute to answering the questions initially posed. Since anatomical, biochemical, developmental, chronobiological, and sleep-related parameters are to be recorded and synthesised, this is a transdisciplinary approach. (14) Due to the heterogeneous evidence level of the available studies, the data are documented suitably and comments are case-related. The results are categorised in Table 1 according to subject group, so that the risk of bias is reduced if comparable data from different author groups are available for the same topic. (15) Not relevant, as the heterogeneous data structure does not allow for a quantitative synthesis. (16) Studies in which only hypotheses were presented, rather than data, were excluded. Critical reviews on specific questions are mentioned explicitly, so that methodologically questionable studies are not inappropriately overrated. (17) The level of evidence of the studies presented can be seen in Table 1. According to this, there are various study types with or without a control group.

On 22 August 2024, the following search strategy in PubMed returned 115 publications: ((melatonin[Title/Abstract]) AND (infant[Title/Abstract])). Only 45 articles were found under “melatonin” AND “infancy”, so the first **115 articles were examined**. After excluding animal studies (16), protocols without their own data on results (2), and a retraction paper that had been reported twice (2), **95 publications remained**.

Of these 95 studies, the following 55 studies were excluded (Figure 1 and Appendix A), [9,10,80,81,82,83,84,85,86,87,88,89,90,91,92,93,94,95,96,97,98,99,100,101,102,103,104,105,106,107,108,109,110,111,112,113,114,115,116,117,118,119,120,121,122,123,124,125,126,127,128,129,130,131,132]:-additional study protocols without data (2),-a report on children with Smith-Magenis syndrome (1),-a study on sleep and breathing disorders in FBXO11 and RAF1 mutations (1),-a report on a 5-year-old boy with tobacco embryopathy, autism spectrum disorder, ADHD, and insomnia (1),-studies on neonatal asphyxia or perinatally acquired brain damage (11),-a study on the general advantages of tryptohan without specific data on infants (1),-studies on treatment of pain (2),-reports regarding prophylaxis of bronchopulmonary dysplasia or later cardiovascular complications or later borderline personality disorders (3),-studies on the effects of maternal obesity on the composition of colostrum (2),-in vitro studies on the activation of phagocytosis by melatonin in colostrum cells respectively on the effects of melatonin on mononuclear cells in umbilical cord blood (2),-a study on the effects of circadian-adapted lighting in a neonatal intensive care unit (1),-studies on postpartum depression or postpartum fatigue in mothers (2),-studies on maternal magnesium deficiency (2),-speculations without measured values about yin-yang models and noradrenaline with references to numerous diseases (1),-speculations without measured values about yin-yang models and noradrenaline with references to stroke and coronary heart disease (1),-theses without a corresponding database on ‘melatonin dysregulation’ as a ‘causal factor’ for autism spectrum and ADHD (1),-theses without a corresponding database on ‘melatonin dysregulation’ as a ‘causal factor’ for sudden infant death syndrome (8),-a study with a healthy control group examining sleep architecture after acute cyanotic apnoea without data on melatonin (1),-theses without data on the influence of calcium, serotonin, and melatonin on infant colic (3),-a report on the diagnosis of cow’s milk allergy (1),-reports on physiology of puberty (2),-a thesis on manic states in mothers (1),-a study on the effect of season on the sleep architecture of infants without estimation of light exposure and melatonin levels (1),-an investigation of diurnal variation in the frequency of the time of delivery (1),-a study on the effect of supplementation of nutrition with tryptophan, adenosine, and uridine in infants aged 8–16 months, cited by Friedman 2018 [95] (1), and-a study with retrospective evaluation of defined geomagnetic activity patterns in temporal correlation with SIDS cases was not considered [131] (1), since this thesis by the same group of authors could not be correlated with structural changes in the pineal gland in animal experiments [132].

A case report on a child with anophthalmia [133] is mentioned in the discussion of this article. The review by Garcia-Patterson et al. 1996 [134] is not presented in Table 1, as the studies cited there by Attanasio et al. 1985 [135] and Waldhauser et al. 1988 [25] are already presented in more detail in the introduction to the present paper and in Table 1. This review discusses some studies on SIDS hypotheses that were not considered in Kennaway’s critical assessments [136,137].

The remaining **38 studies are presented in Table 1, organised by topic**. In addition, the references of the publications listed in Table 1 were checked and further studies were included, which the author has collected, in particular, via PubMed and in patent databases in recent years using the keywords melatonin, infants, newborn, child, pineal gland, noradrenaline, timezyme, and chronutrition [18,22,57,73]. The review published online on 24 April 2024 in Somnologie was based on the following search in PubMed: ((melatonin[title/abstract]) AND (infant[title/abstract])). This generated 108 results, including 10 clinical studies. Of these, two studies were in pregnant women and those with cow’s milk intolerance, two were study protocols, and two were animal experimental or cell biology studies, thus leaving four clinical studies [18]. In the narrative review on the chronobiological peculiarities of early childhood nutrition, 90 literature sources were cited [22]. The narrative review of melatonin metabolism, which was prepared with a clinical pharmacologist and an occupational health physician, contained 51 literature references [57]. The current questions about the toxicity of melatonin in infancy have been discussed in depth with an experienced clinical pharmacologist (see Paditz et al., 2023 [57]).
children-11-01197-t001_Table 1Table 1Studies on melatonin in infancy (PubMed 22 August 2024).FocusAuthor (Year)Type of StudyPostnatal development of the circadian melatonin rhythm in infantsKennaway (2000) [136]**Narrative review** with reference to several studies showing that infants do not exhibit their own circadian melatonin rhythm in the first week after birth [26,138], in the second week [139], or until the third month [140]. In full-term infants, this was not found until the age of 9–15 weeks. A hypothetical dose calculation for oral melatonin administration in infants based on the amount of melatonin administered in breast milk during a night, with reference to the data from Illnerova 1993 [37], is interesting. According to this, infants would require a dose 600 times lower than adults to achieve a night-time melatonin peak.Critical discussion of speculation about unfounded connections between dysfunction of the pineal gland and sudden infant death syndrome or scoliosis.McGraw 1999 [31]**Longitudinal study** (N = 1, documentation “of the interaction between the development of the circadian rhythm of sleep, wakefulness, temperature, melatonin in saliva, and feeding in human infants and the influence of photic and non-photic factors on the initiation of entrainment” for 6 months). “The sleep circadian rhythm appeared last, attaining significance after day 56. …The infant’s nocturnal sleep onset was coupled to sunset before day 60 and subsequently to family bedtime, giving evidence of initial photic entrainment followed by social entrainment”.Carballo 1996 [141]**Cross-sectional study in several age groups** from 3 months to 15 years of age, with determination of melatonin concentration during the day (9:00 a.m. to 9:00 p.m.) and at night (9:00 p.m. to 9:00 a.m.). In the age group 3–18 months, low melatonin concentrations without a clear day–night rhythm (during the day 34.51 ± 8.83 pg/mL, N = 8; at night 43.04 ± 21.9 pg/mL, N = 9). A clear day–night rhythm was only observed in the group aged 18 months to 6 years.Kennaway 1996 [137]**Prospective longitudinal study** on factors associated with development of the circadian rhythm of 6-sulfatoxy-melatonin excretion in urine within 24 h in four time intervals in infants up to 6 months of age (N = 163, including 31 full-term newborns at the age of 55 weeks post-conception and premature infants with five defined risk factors, such as premature rupture of membranes or being a sibling of a sudden infant death syndrome (SIDS) case, and examination of seasonal factors). The circadian melatonin rhythm was found to be programmed to start between the 49th and 52nd post-conception week in full-term infants. At the postnatal age of 6 to 15 weeks, a seven-fold increase in 6-sulfatoxy-melatonin excretion occurred in full-term infants. Perinatal factors were found to be implicated in delaying this development. Premature rupture of the membranes was associated with significant delay in the development of circadian melatonin rhythmicity. Seasonal influences were not detectable. SIDS was not associated with delays in the sibling cohort.Melatonin in human milkItalianer 2020 [70]**Systematic review** (83 reports, 71 human milk components). Eight substances showed significant circadian rhythms (melatonin [20,37,142,143,144,145,146], tryptophan, cholesterol, fats, iron, cortisol, cortisone, and triacylglycerol).Oliveira 2024 [147]**Scoping review** (29 reports, 1993–2023), including 11 studies in which melatonin concentrations were measured during the day and at night. Some of these data are shown in aggregated form in Figure 2 These data suggest that infants from the first day of life until at least 6 months of age receive natural chrononutrition with high melatonin concentrations at night and significantly lower concentrations during the day via the gold standard of breast milk.Akanalci 2024 [148]**Narrative review** (Melatonin, two reports: Qin et al., 2019, Katzer et al., 2016 [38,146]).LactMed 2024 [149]**Short narrative review**. Reference to five studies on melatonin concentration in human breast milk in a day–night comparison (Katzer 2016, Molad 2019, Qin 2019, Biran 2019, Italianer 2020 [38,70,143,146,150]).Jin 2021 [151], Monfort 2021 [152]**Laboratory analysis of melatonin in human milk**. Sample volume of 100 μL used for analysis, overall peak recovery 101.7% with relative standard deviation (RSD) 5.1% [151], stable in breast milk at room temperature for 24 h, at −20 °C for two weeks, and at −80 °C for one month [152].Anderson 2017 [153]**Narrative review** with following statements:-“It should be noted that infants do not show a circadian production of melatonin until they are 3–5 months old”,-“Melatonin should be added to a night-time specific formula feed, in order to bring formula feed closer to the benefits of breast milk [154]. The absence of melatonin in formula feeds is a major deviation from the evolutionary forces that underpin the presence of melatonin in night-time breast milk”,-“Further directions include investigations as to whether: … the addition of melatonin would bring formula feed closer to the benefits of breast milk”.Cohen Engler 2012 [20]**Cohort study** N = 94 mothers, of whom 57% exclusively breastfed and 43% exclusively formula-fed their 2–4 month-old infants, questionnaire on infant colic and sleep disorders in infants) and measurement of melatonin concentration in milk samples from five mothers every 2 h over 24 h, compared with 15 samples from three different types of formula without product information).Less infant colic was observed in breastfed infants (56% vs. 72.5%, *p* < 0.05);Melatonin was not detectable in any of the three types of formula; andMelatonin concentration in breast milk at night was found to be up to 42, and was not detectable during the day (without measurement unit, ELISA test kit)Nabukhotnyi 1991 [155]**Cohort study**. Detection of increased melatonin concentrations in the blood plasma of infants and in early breast milk in the first 3 days of life. Both concentrations decreased between the 5th and 7th day of life. These data indicate that infants depend on melatonin transfer via breast milk in the first week of life.Kivelä 1990 [139]**Longitudinal study** (N = 14–22 mothers and their infants from birth to the 8th day of life). No differences in melatonin concentration in the blood of the mother and in the umbilical cord blood. No differences in the 6-sulfatoxy-melatonin concentration between mother and child during the day (8:00–20:00) and at night (20:00–8:00). A day–night rhythm was not yet detectable in either of them in the first week of life. The melatonin excretion of the newborns was only 1–5% of that of adults, at 2–5 pmol/12 h. Two different melatonin metabolites were detected in the urine of the infants. This could indicate that the neonatal melatonin metabolism is still immature. Overall, the data indicate that the melatonin concentrations of infants immediately after birth reflect maternal melatonin secretion.Chrononutrition for infants and breast milkCaba-Flores 2022 [71] **Narrative review** which contains information on five studies regarding the day–night rhythmicity of melatoninin colostrum [142],in the blood and milk of ten mothers 3–4 days after delivery [37],in the milk of five mothers of healthy 2 to 4-month-old infants every 2 h over the course of 24 h, and examination of the lower incidence of colic attacks (*p* = 0,04), lower severity of irritability attacks (*p* = 0.03), and trend for longer nocturnal sleep duration (*p* = 0.06) in comparison to infants fed with breast milk vs. infant formula [20],in saliva during the development of circadian rhythms in a human infant (N = 1) [31],in urine [156], andreference to the animal experiment-based theory that pineal melatonin synthesis is disrupted after caesarean section as a result of TNF-alpha suppression (chronodisruption) [157].Booker 2022 [158]**Online anonymous survey** (N = 329 mothers). Delayed sleep onset of infants in association with mistimed expressed breast milk vs. direct breastfeed (*p* < 0.001).Wong 2022 [159]**Narrative review** with reference to three studies on infant chrononutrion via breast milk and the underlying day–night rhythm of melatonin concentration in breast milk [20,37,41]; reference to increased sleep disorders [41] and infant colic [20] in non-breastfed infants.McKenna 2018 [160]**Narrative review** providing suggestions for taking circadian rhythms (light, nutrition) into account in neonatal intensive care units.Arslanoglu 2012 [66]**Narrative review**. “WAPM Working Group on Nutrition” report on circadian fluctuations in the concentration of melatonin in breast milk and the day–night rhythm of infant sleep, which can only be observed from the age of four months. Biochemically based recommendations for chrononutrition in infants are derived from this.Aparicio 2007 [65]**Prospective randomised cohort study with three intervention arms**. N = 18 healthy infants, age 12–20 weeks, who had previously been fed with artificial milk. Three study arms, each 1 week in duration: (a) standard commercial infant milk Blemil plus forte, Ordesa with 1.5 g tryptophan/100 g protein, (b) at night 18:00 to 06:00 Blemil, during the day Blemil with 3.4 g tryptophan/100 g protein, (c) during the day Blemil, at night Blemil with 3.4 g tryptophan/100 g protein. Diary method and actigraphy with an actiwatch device utilized, along with night-time urine examination to determine the concentration of noradrenaline, dopa, dopamine, 5-hydroxyindolacetate, 5-HT serotonin, and 5-hydroxy tryptophan. Small but significant improvement in defined sleep parameters in groups b and c compared to a, but no differences between b and c.Concluding comments: “There is wide evidence relating sleep with food intake … First, in human beings, meals serve as Zeitgebers-signals used to synchronize the endogenous clocks, thereby controlling the activity of the circadian regulatory mechanism of sleep (Lohr and Sigmund 1999)[161]”.“These results stress the convenience of developing chronobiologically dissociated diets for infants, and can most probably be extended to components other than tryptophan. In this way, the use of different formulations for day and night constitutes an interesting improvement in artificial infant nutrition which can be termed under the concept of “chrononutrition”. It is interesting to note that the improvement in the sleep of infants was obtained by the simple means of a chronological variation in the normal components of food with no pharmacological intervention”.Cubero 2005 [156]**Prospective cohort study** (N = 16 infants of 12 weeks of age undergoing natural or artificial feeding, sleep parameters measured by actimeter for a week. Detection of circadian rhythm of 6-sulfatoxy-melatonin in urine, and in breast milk tryptophan was measured). Acrophase of 6-sulfatoxy-melatonin at 06:00 in the breastfed infants, and at 03:00 for tryptophan; assumed sleep, actual sleep, and sleep efficiency were significantly increased in the breastfed infants as compared to the formula-fed infants.McMillen 1993 [162]**Longitudinal cohort study** (N = 23 full-term and 22 preterm infants and their mothers, sleep diary for mother and child and 24 h profile of melatonin and cortisol in the mothers’ saliva in the 2nd to 10th week postnatally). Full-term babies found to sleep better than preterm infants; the mothers of preterm infants slept worse and had significantly lower nocturnal melatonin concentrations in their saliva (discussed as a “greater physiological disruption”). It can therefore be assumed that preterm infants receive less melatonin from their mothers’ milk than full-term infants.Nocturnal melatonin synthesis in infants aged 4, 6, and 9 monthsFerber 2011 [163]**Prospective cohort study** (N = 44 infants, night-time urine 19:00 to 07:00 for determining 6-sulfatoxy-melatonin at the ages of 4, 6, and 9 months, neurobehavioral Assessment of Preterm Infant Behaviour (APIB) at 2 weeks postterm age, and mental developmental index Bayley Scales of Infant Development (Baley-II) at 4, 6, and 9 months): Low melatonin concentrations of 10 and 14 μg in premature and term births at the age of 4 months, medium values of 20 and 38 μg at the age of 6 months, and high values of 40 and 55 μg at the age of 9 months;Significant correlations between neuropsychological parameters (APIB) and age-related melatonin production at the ages of 6 and 9 months.The authors’ conclusion: “Because melatonin synthesis is affected by noradrenergic stimulation, the irregularity in sympathetic activity in preterm infants may affect the maturation of melatonin production. This is buttressed by the results of the current study, which shows long-term correlations between the integrity of the autonomic system at 2 weeks corrected age and the melatonin levels up to 9 months. The compromised noradrenergic effect on melatonin production on one hand, and the resulting lack of melatonin facilitation for reduction of oxidative stress on the other, may be a bidirectional process affecting sympathetic stability in preterm infants. Such a bidirectional coregulatory paradigm for explaining processes in the developing brain was suggested.”Melatonin receptors (MR) and incidence of sleep disorders in infantsLin X 2022 [164]**Prospective cohort study** (effects of maternal and father´s sleep and emotions on infant sleep) **and case-control study** (methylation of promoter regions for MR). Incidence of sleep disorders in infants 0–3 months was found to be 40.5% (N = 513, analysed by Brief Screening Questionnaire for Infant Sleep Problems, BISQ [165]),** associated with decreased MR expression by up-regulating melatonin receptor 1B gene (MTNR1B, chromosome 11q14.3) methylation.Sulkava 2020 [166]**Prospective cohort study** (N = 1301). Evidence found supporting a genetically determined reduced number of type 1 melatonin receptors in 8-month-old infants in connection with a variant of the melatonin receptor gene MTNR1A on chromosome 4q.Melatonin use in infants and toddlersOwens 2024 [58]**Online survey** (N = 3063 caregivers). According to this, melatonin is given to 1.7% of children aged 0–36 months in a largely uncontrolled manner.Melatonin and sleep in infantile epileptic spasms syndrome (IESS)Sun 2024 [167]**Randomized, placebo-controlled, double-blind trial (RCT)**. N = 35 with melatonin 3 mg administered between 20:00 and 21:00 daily vs. N = 35 administered a placebo for 2 weeks, 0.5–1 h before bedtime, age 3 months to 2 years. Serum melatonin level at 06:00 h 84.8 vs. 17.5 pg/mL (*p* < 0.001), improved sleep quality 85.7% vs. 42.9%, (*p* < 0.01); Infant Sleep Assessment Score (ISAS) in 4–11-month-old patients 29.3 vs. 35.2 (*p* < 0.01), shortened sleep-onset latency 6.0 vs. 3.0 min (*p* = 0.030); no significant improvement in seizures with simultaneous administration of anticonvulsant therapy with adrenocorticotrophic hormone (ACTH) and magnesium sulfate (MgSO_4_).Melatonin and neonatal pain in premature newbornsSanchez-Borja 2024 [168]N = 61 non-hypoxic preterm infants; reduced melatonin concentration in plasma on the third day of life correlates with more severe pain (Premature Infant Pain Profile = PIPP score > 5, *p* = 0.03) ***Effect of pasteurisation of donor breast milk on melatonin concentrationBooker 2023 (1) [169] Holder pasteurization of night-time milk samples (N = 10) from donors to a temperature of +62 degrees Celsius for 30 min reduced melatonin concentration (51.92 vs. 39.66 pg/mL (*p* < 0.01).Booker 2023 (2) [170]Rapid or slow cooling of night-time milk samples (N = 27) from donors to a temperature of +4 degrees Celsius reduced the melatonin concentration. After pasteurisation, the melatonin concentration in the milk remained stable.Melatonin and sudden death or apparent life-threatening event (ALTE)Bishop-F. 2022 [55], Shimomura 2019 [171]**N = 6 and N = 1 deaths** at the age of 2 months to 3 years, melatonin concentrations 3 to 1400 ng/mL in post-mortem blood due to uncontrolled overdose.Labay 2019 [56] **N = 2 deaths** at the age of 9 or 13 months in connection with co-sleeping of twins or tobacco exposure, melatonin in blood 13 ng/mL and in gastric fluid 1200 ng/mL or 210 ng/mL in blood, respectively.Gitto 2011 [172]**Narrative review**. “After birth, the full-term neonate does not exhibit a day:night melatonin rhythm for 2–4 months, leading to transient melatonin deficiency [21,28,29,137,173,174]. At birth, Munoz-Hoyos et al. … observed the absence of a circadian rhythm of melatonin during the early neonatal period [175]. Premature delivery of the newborn leads to a more prolonged relative melatonin deficiency. Moreover, the onset of pineal melatonin secretion is even more delayed with the occurrence of neurologic insults [137,174]. Obviously, in both normal and neurologically damaged premature neonates, the melatonin deficiency persists for a longer period [28,137,173,174]. Thus, an infant born 3–4 months prematurely may lack significant melatonin levels for 7–8 months or longer.” (Note: some studies make questionable suggestions regarding a link between ALTE or SIDS and pineal immaturity. These studies are not discussed again here, as they have been discussed and criticised in detail by Kennaway 1996 and 2000 [136,137].)Sivan 2000 [176]**Case-control study with four arms** (N = 80 infants at post-conceptional age of 48–58 weeks, determination of 6-sulphatoxy-melatonin in urine with the following groups: (a) N = 35 healthy age-matched control group to groups b, c, and d, (b) N = 15 ALTE, (c) N = 15 after cyonotic apnoea episodes requiring mouth-to-mouth resuscitation, (d) N = 15 siblings of SIDS cases). Melatonin in urine after ALTE was significantly reduced compared to controls; no differences between groups a, c, and d.Sturner 1990 [177]**Autopsy data SIDS vs. control group**. Data from 68 infants whose death was attributed to either SIDS (N = 32, 0.5–5.0 months old; mean +/− SEM, 2.6 +/− 0.2 months) or other causes (non-SIDS, n = 36, 0.3–8.0 months old, 4.3 +/− 0.3 months). Measurement of melatonin concentration in whole blood, ventricular cerebrospinal fluid (CSF), and/or vitreous humour (VH). Significant correlation found between melatonin concentrations in different body fluids of the same individual. After adjustment for age differences, the melatonin level in the cerebrospinal fluid was significantly lower in SIDS children (91 +/− 29 pmol/L, n = 32) than in children who died for other reasons (180 +/− 27; n = 35, *p* < 0.05). “These differences did not appear to be explained by the interval between death and autopsy, gender, premortem infection, or therapeutic measures initiated before death. A reduced melatonin production could possibly indicate a disruption of the maturation of the physiological circadian organisation in association with SIDS”. (For a critical commentary on this study, see Kennaway 1996 and 2000 [136,137].)BasicsHeine 1995 and 1999 [52,53]Heine has demonstrated that the pineal synthesis of melatonin in infants is only possible after the successful enteral absorption of tryptophan as the key precursor. This means that a sufficient amount of alpha-lactalbumin is required as the transport protein in blood. In addition, there is significant competition between tryptophan and neutral long-chain amino acids (LNAA) for passage through the blood-brain barrier (BBB). Therefore, a defined concentration ratio between tryptophan and LNAA must be present, which should be taken into account in the production of formula nutrition.For more details, see the results in this review; (**) Definition of sleep disturbance in infants according to BISQ: waking up three times or a waking duration of more than 1 h or a total sleep duration of less than 9 h per 24 h [164,165]; (***) A similar RCT by Behura et al. (2022) was not considered because the associated melatonin concentrations were not measured [96].


## 3. Results

The publications listed in PubMed were analysed based on the groups of subjects in Table 1. This accumulated knowledge surrounding several detailed questions provides consistent data showing that the pineal gland does not express rhythmic melatonin synthesis and secretion after birth until at least the third to fourth month of life. In premature births and especially after premature rupture of the membranes, this development may be even more delayed. In the first few months of life, infants are therefore dependent on external melatonin supply via breast milk. Donor milk or industrially produced formula feeding can only imitate the day–night rhythm of melatonin supply for infants if a differentiation is made between melatonin-poor day milk and melatonin-rich night milk. The nature of infant nutrition is, from an evolutionary biological perspective, designed for circadian-oriented ‘chrononutrition’ in all mammals (Figure 2). In practice, this requires that non-pooled milk be used for breastfeeding, for collecting donor milk, and for the industrial production of infant formula.

The following considerations help to understand why the pineal gland in infants is only able to ensure its own melatonin synthesis and secretion after several months.

From an epistemological point of view, it seems important to recall the paths and pitfalls that have led to current ideas and results, or blocked them for several centuries, before presenting current molecular genetics and developmental biological data:-In 1898, Otto Heubner reported on a 4.5-year-old boy with precocious puberty and tall stature, in whom a pineal tumour was detected at autopsy [180]. This case report drew attention to the endocrinological functions of the pineal gland. The discovery of melatonin by Lerner in 1958/1959 [181,182] marked the beginning of an exponential increase in knowledge about melatonin, which continues to this day.-In 1924, Ladislaus von Meduna (1896–1964) submitted a fundamental histological study on “the development of the pineal gland in infancy”, in which anatomical preparations of 30 pineal glands from children from the neonatal period to the age of 4 (N = 26) and after the age of 4 (N = 4) were examined [183]. These studies clearly show that the microstructure of the pineal gland develops and differentiates in several phases only within the first year of life, so that its functional capacity is not established until several months after birth.

The vascularisation of the pineal gland occurs from the third month of the foetal period [183] (p. 535). After birth, the pineal gland would consist of “neutral ectodermal cells” until the 2nd to 4th month of life, and their differentiation into plasmatic and fibrous glial cells or into pineal cells (pinealocytes) does not occur until between the 3rd and 8th month [183] (p. 546). From today’s perspective, von Meduna described 100 years ago that myelinisation processes, which are a prerequisite for the transmission of information via nerve fibres in the pineal gland, only start between the 3rd and 8th month after birth:

“Typical astrocytes [as a subgroup of glial cells] in a spider-like shape with many fine processes... often condense around the nucleus in a membrane-like manner” [183] (p. 542). In the 7th to 8th month of life, “the pineal gland would show the most vivid picture” because the pineal cells now develop numerous processes “to reach the nearest vessel or septum” [183] (p. 544).

In a third developmental step of the pineal cells, there are bulb- or pear-shaped extensions of fibres of the pineal cells (“end bulbs”), which connect with a “dense network of strong neuroglial fibres” [183] (p. 547).

Meduna suspected that the pineal gland was an endocrine organ [183] (p. 547). The postnatal differentiation processes of the cells of the pineal gland described by Meduna were confirmed in 1987 by Min et al. using immunohistochemical methods on 16 pineal glands from infants and toddlers aged 38 weeks of gestation or older up to 3 years of age. In the neonatal period, predominantly pigmented type I cells were detectable, in which no neuron-specific enolase was detectable. At the end of the first year of life, type II cells with positive neuron-specific enolase then dominated [184]. These data indicate that the innervation of the pineal gland develops only postnatally, since neuron-specific enolase is considered a marker of neural maturation [185]. The importance of microglia in the postnatal networking of pineal structures (cells, nerves, blood vessels) has been extensively documented, at least in Wistar rats [186].

The mass of the pineal gland of 36 Merino sheep increased with postnatal age when compared between the 1st and 6th and 9th and 24th months after birth (54 vs. 66 mg during the day and 63 vs. 77 mg at night). At the same time, higher melatonin concentrations were measured at night in the 9th and 24th months [187] (Table 1 and Table 2). A similar increase in size and cellular differentiation was also observed in goats up to the age of 4 months [188]. In Wistar rats, cAMP signals triggered by noradrenaline to activate pineal melatonin synthesis also only become maximally effective 15 days after birth (Figure 3) [189]. In humans, this would correspond to a postnatal age of about 4.3 months [190].
children-11-01197-t002_Table 2Table 2Melatonin concentrations in the serum of newborns.Autor (Year)SettingResultsAssessmentBülbul (2024) [191]N = 35, birth weight 3321 ± 474 g, gestational age 38.1 ± 1 weeks, spontaneous birth 37.2% (13/35), Caesarean section 62.8% (22/35), gemale 60% (21/35), babies stayed with their mothers, room light 6–10 lux.Serum melatonin at 2:00 a.m. (pg/mL) 19.9 ± 4.38 (9.9–26.3).No information on the age of the babies. It can be assumed that they received breast milk (for comparative data, see above in the text).Muñoz-Hoyos (2007) [192]N = 35, birth weight 1800 g (870–4400 g), gestational age 32.5 (26–40) weeks, with respiratory distress syndrome, without sepsis, light: 300–450 lux in the morning.Serum melatonin at 9:00 a.m. in the group > 1500 g on the 1st and 7th day 104.2 ± 22.9 and 109.4 ± 24.0 pg/mL; in the group < 1500 g on the 1st and 7th day 63.2 ± 6.2 and 79.3 ± 6.8 pg/mL (*p* = 0.017).No information on diet, so it can be assumed that the babies received breast milk. Significantly lower melatonin concentrations in the group with a weight < 1500 g.Sánchez-Borja (2024) [168]N = 61 preterm infants < 25, birth weight 1350 (800–2055 g), gestational age 29.9 (24–34) weeks, 65.6% (40/61) with parenteral nutrition.Serum melatonin on the 3rd day of life between 8:00 and 9:00 a.m. 30.6 (12.3–76.6) pg/mL.No information on oral feeding.


It has been confirmed that the pineal gland consists of 90–95% pinealocytes, in which pulsatile pineal melatonin synthesis takes place [193,194]. More than 5000 cells of the pineal gland were examined using single cell analysis. As a result, six cell types were found: pinealocytes, astrocytes, microglia, vascular, epithelial and leptomeningeal cells, [60], to which a whole series of marker genes have been assigned [195] (Figure 2 in the citated paper).

In 2023, Gregory et al. reminded us of the innervation of the pineal gland by the nervous conarii [196,197], which, as a non-myelinated postganglionic sympathetic nerve, is an essential part of the signal transduction chain between theretina, SCN, superior cervical ganglion (SCG), and glandula pinealis (Figure 4).

Hertz et al. investigated the influence of these nerve fibres on Lhx4 expression in rats after SCG ganglionectomy using radiochemical in situ hybridisation [198]. The interruption of the nervus conarii eliminated the pineal Lhx4 expression that occurs in darkness. This could be stimulated again pharmacologically using isoprenaline. These data suggest that Lhx4 and noradrenaline are involved in the activation of melatonin synthesis and thus also in the activation of timezyme via the coronary nerve [198]. However, there also remain unanswered questions; extensive gene analyses have provided evidence for two pathways (phototransduction pathway and aldosterone synthesis and secretion pathway) that do not focus solely on Lhx4. However, Lhx4 can be assigned to the phototransduction pathway [198] (p. 7). Figure 5 shows that the size of the pineal gland only increases significantly after birth. At the same time, Lhx4 becomes increasingly detectable (Figure 5).

Lhx4 is a protein that is encoded in humans by the LHX4 gene, with six exons on chromosome 1q24.1–1q24.3 [199], and which, based on the above-mentioned results, is significantly involved in the control of the differentiation and development of the pineal gland. The LIM homeobox 4 (Lhx4) genes contribute to the regulation of melatonin synthesis in pinealocytes [200]. LIM genes were discovered first in developing tissue types of the pituitary gland, retina, thymus, limbs, pancreatic islet cells, spinal cord, and brain [201]. LIM is an acronym that refers to the associated homeodomain proteins described first (Lin-11, Isl-1, Mec-3) [201]. LIM proteins are important for organic and neuronal development processes [202]. In terms of classification, LIM domains are part of the more than 6500 known zinc finger domains. Zinc-containing polypeptides support the structural and functional flexibility of the 20 canonic amino acids found in humans [203].
Figure 4Representation of the signal transduction chain for the activation of pulsatile pineal synthesis and secretion, with indication of the localisation of M1 and M2 melatonin receptors in the brain that are associated with sleep [204,205]. Light is converted into chemical impulses in the photosensitive ganglion cells of the retina, which stimulate the retinal formation of melanopsin. Melanopsin inhibits the synthesis of melatonin [206]. The central master clock SCN is activated via the retinohypothalamic tract. Via several sympathetic ganglia (paraventricular nucleus (PVN), upper thoracic medulla, cervical ganglion), melatonin synthesis in the pineal gland is inhibited by light or activated by darkness in the evening. The pineal gland secretes melatonin directly into the cerebrospinal fluid and via venous effluents into the jugular vein. Melatonin exerts its effects in particular via two receptor types (M1, M2), which stimulate the switch from wakefulness to sleep in the frontal pre-cortex (M1), after melatonin has induced the transition from wakefulness to NREM sleep via feedback mechanisms to the SCN (M1, M2). The thalamus, as the ‘gateway to consciousness’, is sent into NREM sleep via M2 receptors and is opened in this state for the transfer of verbal information from short-term memory to long-term memory in the hippocampus; the consolidation of memory content takes place to a large extent during undisturbed sleep. The transition from NREM to REM sleep is induced by M2 receptors in the ventrolateral periaqueductal grey matter. During REM sleep, REM muscle atonia is generated via several neural switching points, and motor information can now be stored (‘You learn to ride a bike in your sleep’). The basal forebrain is involved in these processes (M2). A highly simplified overview based on [79,204,207,208]. Slightly modified according to Paditz [209], with kind permission.
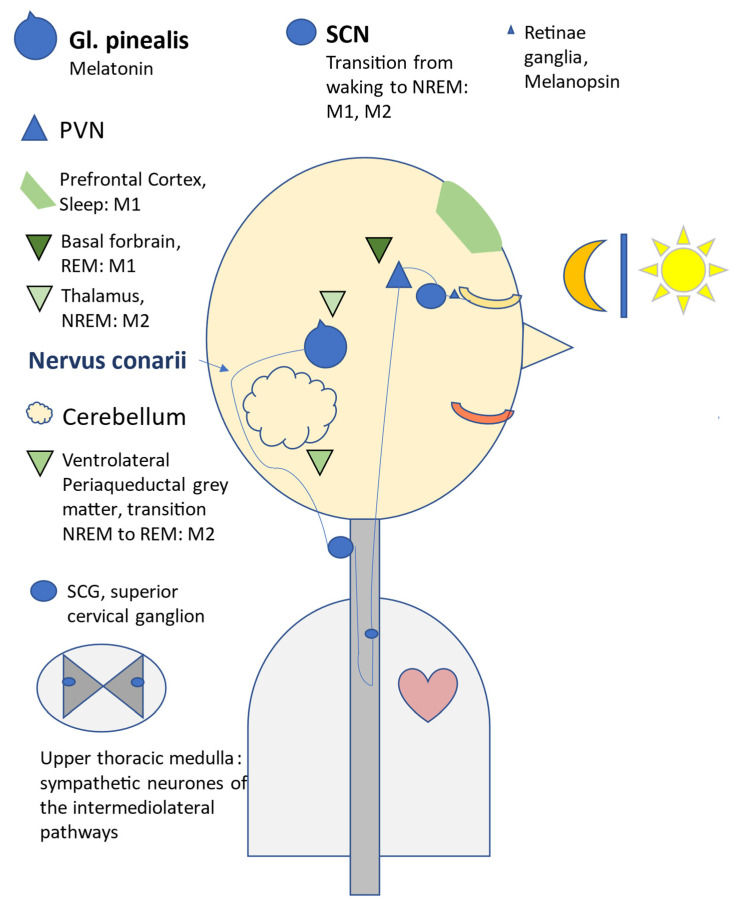

Figure 5Expression of Lhx4 in the developing rat pineal gland. The arrow points to the pineal gland. Scale bar, 1 mm; E, embryonic day; P, postnatal day; ZT, zeitgeber time. Radiochemical in situ hybridisation for detection of Lhx4 mRNA in coronal sections of the brain from rats sacrificed at ZT6 (**left**) and ZT18 (**middle**) at the indicated developmental stages (**one per row**) ranging from E15 to P30. ZT18 sections were counterstained in cresyl violet for comparison (**right**). Reproduced from Hertz [198], with kind permission.
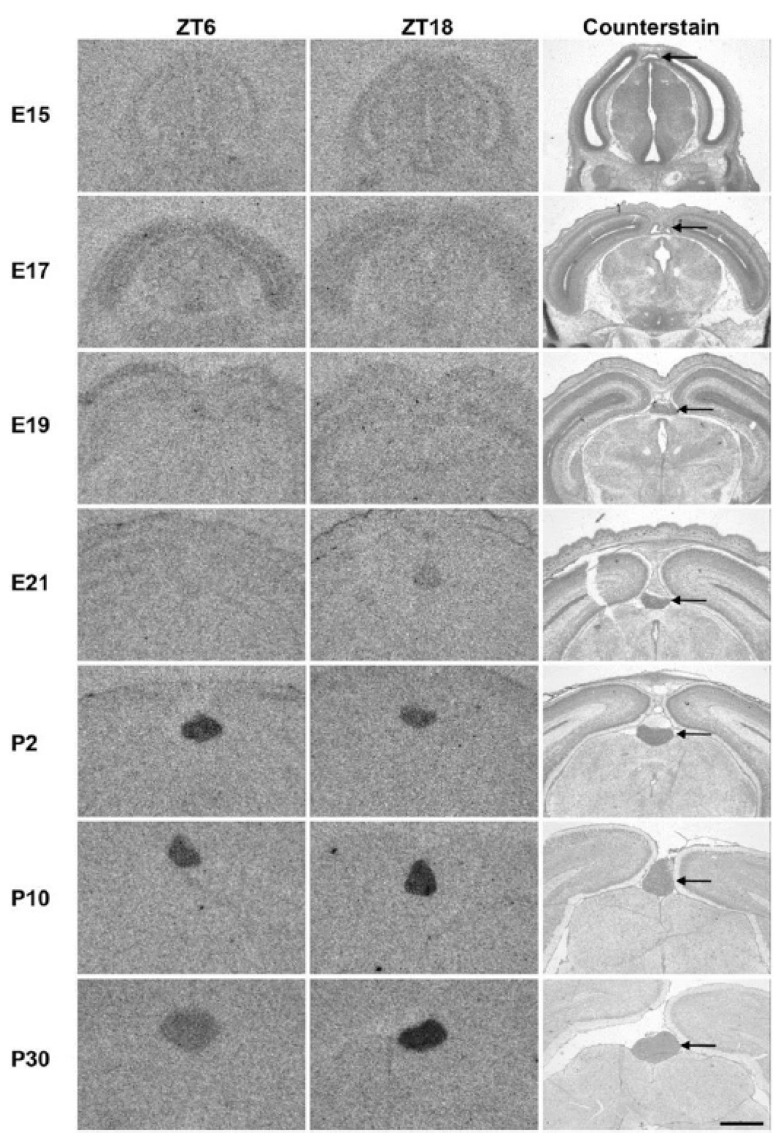



## 4. Discussion

The data presented here show that there is still no systematic review regarding why the human pineal gland does not exhibit its own melatonin synthesis and secretion in the first months of life, including the circadian rhythm related to the pineal gland. This review is intended to fill this gap.

The initial hypothesis that infants do not have their own pulsatile pineal synthesis and secretion in the first days and months of life is not called into question by the studies listed in Table 2. Rather, it can be assumed that the low nocturnal melatonin concentrations in newborns listed in Table 2 originate from prenatal maternal and placental sources [75,210,211]. In pregnant women, for example, melatonin concentrations in saliva of 23–25 pmol/L (=5.35–5.81 pg/mL) were measured at 3:00 a.m., regardless of the duration of pregnancy [212] (Figure 1 in the citated paper). Immediately after birth, mean melatonin concentrations of 36.8, 23.8, and 32.7 pg/mL were found, respectively [39]. (see also [22] (Table 1 and Figure 8 in the citated paper) and [18] (Figure 2 in the citated paper). Qin et al. found comparable melatonin concentrations of 23.5 pg/mL in breast milk at 3:00 a.m. during the first 30 days [38].

In adults aged 25.9 ± 4.7 (20–32) years, maximum nocturnal melatonin concentrations of 101.1 ± 3.5 (18–163) pg/mL were reported. At the age of 59 ± 10.0 (49–73) years, these concentrations were lower at 49.4 ± 38.1 (14–150) pg/mL [213].

It should also be noted that the half-life of elimination is significantly longer in premature babies than in young adults (6.20 to 21.02 h vs. 53.7 ± 7.0 min) [62,63], see also [18] (Table 2).

Based on the above-mentioned histologically detectable developmental and differentiation steps described by Meduna [183] and Min in infants, and the same in animal studies [189,190,191], it can be assumed that the cause of the delayed development of the ability of the pineal gland to synthesise melatonin in the first months of life lies in the development of the pineal gland itself, which only begins postnatally. Since animal data cannot be extrapolated to humans without further investigation, further studies are needed to clarify how postnatal differentiation and connection of the pinealocytes with the associated structures in the signal transduction chain from the retina to the pineal gland occurs in humans. Lhx4, noradrenaline, and timezyme are likely to continue to play a leading role in this.

From an evolutionary and epistemological perspective, it is interesting that the main human rhythm generators, such as the respiratory centre with the pre-Bötzinger complex [78,214,215,216], the SCN [217], and the pineal gland [198], are controlled and developed by extremely different groups of genes and pathways. According to current knowledge, however, the pineal gland does not have its own rhythm generator, but is dependent on stimulation via the SCN and noradrenaline [198]. Similarly, in one case, a child with anophthalmia did not develop a melatonin rhythm until the child was 9 years old, whereas a circadian rhythm of heart and respiratory rate was already detectable at 1, 8, and 12 months of age [133]. McGraw and Preyer have verified these different maturation times in healthy infants, with the immediate postnatal onset of rhythmogenesis of respiratory and heart rates and temperature regulation, and the onset of circadian rhythmicity of sleep only after several months [31,36].

From a chronobiological perspective, there are three possible solutions for providing infants with sufficient melatonin without external additives in the first 3–6 months:

**1. Promotion of breastfeeding.** In this case, it should be ensured that the mother’s milk contains hardly any melatonin during the day and that the “night milk” contains high melatonin concentrations. If milk is pumped and collected, care should be taken to collect the mother’s milk in different containers, distinguishing between day milk and night milk [22]. This challenge has been pointed out by several groups of authors [18,20,22,66,68,71]. Current studies, such as Häusler et al. [75], are addressing this topic.

**2. Non-pooling of breast milk in breast milk collection centres.** This simple principle should also be taken into account in breast milk collection centres [22]. Studies are also being prepared for this purpose (Erler and Paditz et al.).

**3. Production of formula milk.** The same circadian rhythm can be detected in cow’s milk as in human milk [67,218,219,220,221,222,223]. Therefore, it is suggested that non-pooled cow’s milk be used to produce night milk, which is rich in melatonin, and day milk, which is low in melatonin. The entire production chain, from the lighting of the animal stalls during the day and darkness at night-time, to the times when day milk or night milk should be milked and collected, and to the consideration of pH values and temperatures in the production of milk powder and infant formula, has been registered in a group of patents [18,22,73], so that chrononutrition in accordance with scientific and regulatory standards with optimal melatonin supply for infants without artificial melatonin supplementation is possible. It remains to be seen whether specific AANAT genotypes associated with increased melatonin concentrations will become established in dairy cows [224].

These conclusions are supported by the data presented here. See:-the comments by Aparicio et al., 2007 [65], which have been included in Table 1 in the chrononutrition category as a result of a prospective randomised study (demand for day and night milk for infants and introduction of the term ‘chrononutrition´ for this age group);-the results of the prospective cohort study by Ferber et al., 2011 [163], which showed that attention should not only be paid to the melatonin concentrations in day and night milk, but that maturation processes also correlate with noradrenergic effects on pineal melatonin synthesis and neuropsychological developmental parameters up to the ninth month of life (Table 1, subject group on nocturnal melatonin synthesis);-the results of the longitudinal study by McMillen et al., 1993 [162], which showed a correlation between circadian rhythms and infant maturity and the fact that there is a ‘greater physiological disruption’ of these circadian rhythms in premature infants;-the results of the intervention studies with control groups by Cubero et al., 2006/2007 [40,41], which were reported in the introduction of the present systematic review, showed that in infants aged 4 to 20 weeks, sleep could be improved by supplementation with tryptophan as a precursor for melatonin formation [40,41]. The disadvantages of artificial tryptophan supplements have already been mentioned in the introduction. This does not change the orientation towards the evolutionary establishment of chrononutrition with human breast milk as the gold standard, which should serve as a basis for breastfeeding mothers, human milk banks, and infant formula manufacturers.

Kennaway’s suggestion of low melatonin dosages in infants (Table 1, Postnatal development of the circadian melatonin rhythm) [136] is supported by Hardeland’s orientation towards low dosages. Hardeland referred to low doses, which are necessary to saturate the melatonin receptors in order to induce circadian effects [6]. From a clinical point of view, it is important to distinguish between low dosages for circadian effects and the sometimes higher dosages for antioxidant effects [6]. In addition, it must be noted that a 20-fold longer elimination half-life for melatonin has been demonstrated in premature infants compared to young adults [18,62,63,225], so that cumulative effects are to be expected at repeated doses. To my knowledge, there are a lack of studies to date that allow us to categorise the administration of melatonin during breastfeeding as purely ‘oral’ or whether a significant proportion is absorbed via the oral mucosa, bypassing the liver first pass [22]. In animal experiments, it has been shown that the pineal gland, as a circumventricular organ, first secretes pineal melatonin into the cerebrospinal fluid, so that the brain is supplied with significantly higher melatonin concentrations. In animal experiments, the melatonin concentrations measured in vivo in the CSF were four times higher than the concentrations measured in the blood [22,226,227]. In my opinion, these in vivo data are more significant than the post-mortem measurements of melatonin concentrations in the blood and CSF reported by Sturner et al. (Table 1, melatonin and sudden death) [177].

In the studies on laboratory analysis by Jin 2021 and Monfort 2021, the high stability of melatonin at room temperature and at temperatures of −20 and −80 °C was noted (Table 1, Melatonin in human milk) [151,152], while Booker et al. 2023 found a concentration reduction when cooled to +4 °C. When heated to +62 °C *for 30 min*, a slight decrease in concentration from approx. 52 pg/mL to 40 pg/mL was reported (Table 1, Effect of pasteurisation) [169,170]. To achieve antiviral effects, other authors recommend *short-term heat inactivation* at 62 °C *for 5 s* [228,229,230]. The current state of knowledge on this topic cannot be comprehensively presented and discussed here. In this regard, more detailed data can be found in the review by Oliveira et al., 2024 and in the review by Kilic-Akyilmaz et al., 2022 [147,231]. Since melatonin is also contained in bee honey [232], among other products, it seems important to me to point out that since 1980, there have been several reports of life-threatening or lethal courses of infant botulism in connection with spores of Clostridia in honey (Arnon 1980, and several confirmatory studies cited in Bamumin 2023 [233,234]). The administration of honey to infants is therefore not recommended, as multiple intestinal functions are not yet mature in infancy.

The present systematic review is based on several studies with a high level of evidence, such as Carballo et al., 1996, Kennaway et al., 1996, Aparicio et al. 2007, McMillen 1993, Ferber et al. 2011, Sun et al. 2024, and Sivan 2000 (Table 1), and on several reviews whose authors have made plausible fundamental conclusions (Kennaway 2000, Anderson 2017, see Table 1). Overall, the current state of knowledge regarding the necessity of using melatonin-poor day milk and melatonin-rich night milk as this relates to chrononutrition adapted to the circadian rhythm for infants is based on a compact amount of data. In my opinion, the different quality of the studies available so far is widely compensated by the consistent tenor of the majority of the studies. Open questions have been addressed. The exclusion of studies found in the research performed on 22 August 2024 has been documented for each study (see in particular Appendix A), so that a bias with regard to the selection of studies has been minimised. The PRISMA checklist, which has been expanded to include 23 items [235], was applied in the present review additionally (Appendix A). This seems justifiable to me, since essential questions regarding the transparency and quality of the studies used were also addressed with the 17-item checklist used here. The GRADE criteria (Grading of Recommendations, Assessment, Development, and Evaluations) have been taken into account in Table 1, and in the introduction and discussion of this publication. This also applies to the PICO criteria (participants, interventions, comparators, and outcomes; see introduction and methodology of this paper).

Further research is needed to examine the following open questions in more detail: Which other genetic programmes play a role in the maturation of the pineal gland’s functions after birth? How can the already known and well-documented knowledge about increasing melatonin concentrations in cows’ night milk be taken into account in practice by specifically improving the lighting during the day and at night in cattle sheds? How can the milking practices of milk producers be changed to produce melatonin-rich night milk that would allow the industrial production of chrononutrition for infants? What communication strategies are effective in educating breastfeeding mothers that milk should not be pooled? This challenge also applies to breast milk collection centres. Laughter, positive thinking, restful sleep, and sensible light exposure can increase the melatonin content of breast milk, so this knowledge should be integrated into communication and support for the parents of infants. Before clinical conclusions are drawn from the open questions addressed here, the critical remarks published by Boutin, Kennaway, and Jockers regarding the interpretation of results in relation to melatonin [236,237] should be repeatedly reviewed and taken into account. This standard has been applied to the present systematic review: since initially published associations between sleep disorders in infants were reported, the quality of these findings was then evaluated with the results of placebo-controlled intervention studies, so that qualitative and quantitative relationships could then be assumed.

**Conclusions:** The pineal gland is anatomically present in humans at birth, but is not able to synthesise and secrete melatonin until 4 to 6 months of age. In premature infants, as well as after premature rupture of membranes or caesarean section, this physiological melatonin deficiency in infants may last even longer. The following factors contribute to this physiological melatonin deficiency: 

1. the development and maturation of the pinealocytes, in which the pulsatile circadian melatonin synthesis of the pineal gland takes place, only occurs postnatally in the first months of life; 

2. the innervation of the pineal gland with noradrenergic nerve fibres only occurs after birth in the first months of life; and

3. neurogenetic data indicate that postnatal maturation of the pineal gland is subject to genetic programmes, which, according to current knowledge, include in particular the Lhx4 system.

Physiological melatonin deficiency in the first months of life is associated with sleep disorders, increased crying, infant colic, and possibly also with an increased risk of non-accidental trauma (shaken baby syndrome). These problems are observed less frequently in breastfed babies than in babies fed on formula. The reason for this is that breast milk provides ideal chrononutrition, with low melatonin levels during the day and high melatonin concentrations at night. It is therefore possible to speak of melatonin-rich night milk and melatonin-poor day milk.

The gold standard for the composition of industrially produced infant formula is human breast milk. In terms of melatonin-rich night milk and melatonin-poor day milk, formula milk can be adapted to the gold standard of breast milk without artificial addition of melatonin. This is a significant step on the way ‘back to nature’ if non-pooled cow’s milk is used as a raw material for the production of infant formula. This is because cow’s milk, like the milk of other mammals, also contains more melatonin at night than during the day [218,220,223]. During breastfeeding and in the collection of donor milk in milk banks, this day–night rhythm should also be considered by collecting and administering day milk and night milk separately (non-pooled milk) [18,22,73,238].

## Figures and Tables

**Figure 1 children-11-01197-f001:**
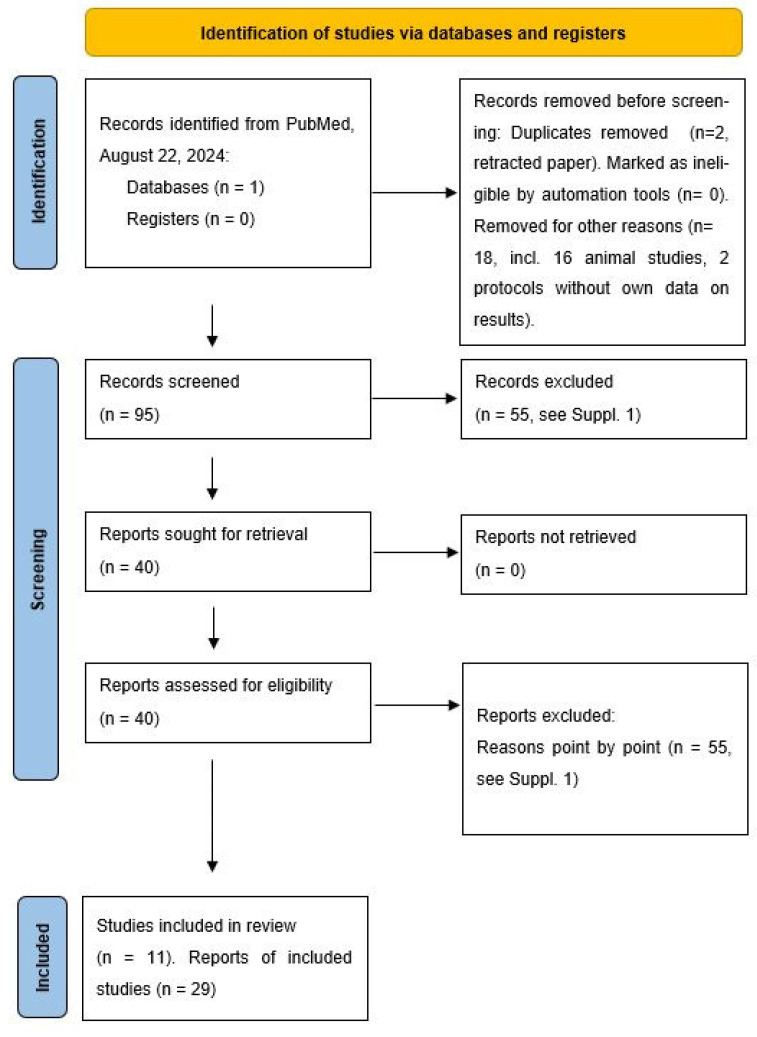
Flow chart with declaration of the selection procedure applied here. List of excluded items, see Appendix A. Source of the flow chart: Page MJ, et al. BMJ 2021;372:n71. doi: 10.1136/bmj.n71. This work is licensed under CC BY 4.0.

**Figure 2 children-11-01197-f002:**
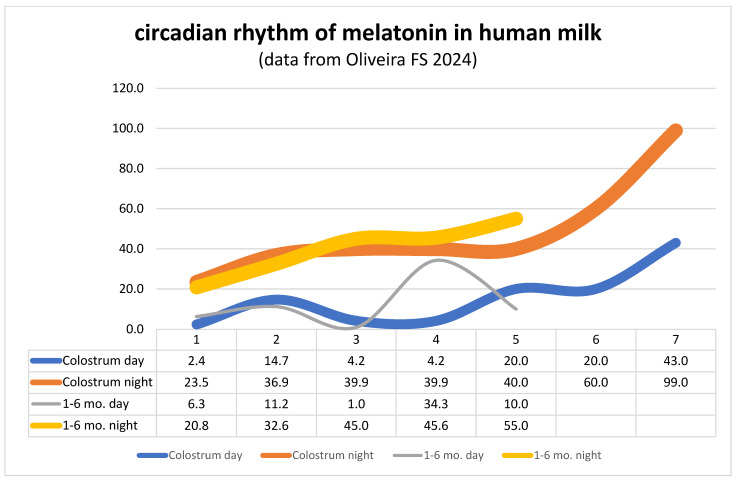
Melatonin concentrations in human breast milk with circadian rhythmicity in colostrum (daytime = blue) and in the first six months (daytime = grey) and at night in colostrum (orange) and in the first six months (yellow). Melatonin concentration in pg/mL. The data were taken from the review by Oliveira et al., 2024 and are presented graphically in aggregate form. The individual measured values for colostrum are taken from the studies by Qin (1), Aparici-Gonzalo (2), Pontes 2007 (3), Pontes 2006 (4), Honori-Franca (5), Silva (6), and Illnerova (7), as well as for mature milk from Kimata (1), Aparici-Gonzalo (2), Cohen Engler (3), Molad (4), and Silva (5) [20,37,38,39,142,143,144,145,147,178,179].

**Figure 3 children-11-01197-f003:**
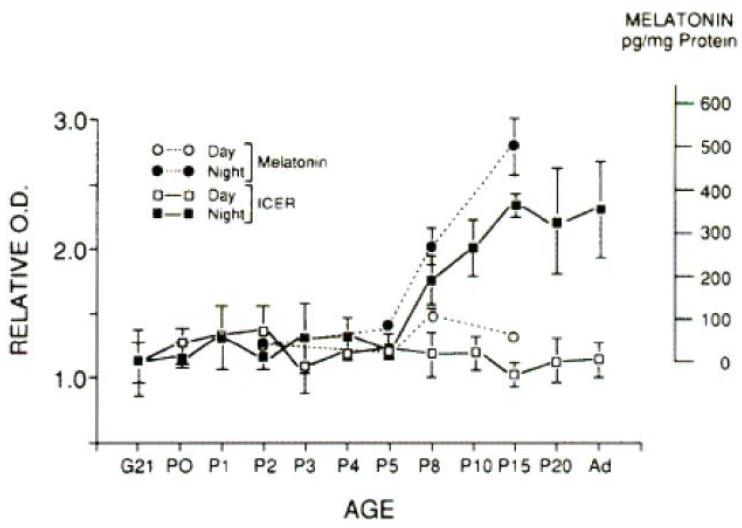
Analysis of relative optical density (O.D.) of inducible cAMP early repressor (ICER) hybridization signal in rat pineal gland (solid line) and superimposed pineal melatonin values (dashed line). ICER night-time values start to be significantly different from daytime values from P8 onward (P8: *p* < 0.05; P10, P15, P20, adult (Ad): *p* < 0.01). Night-time melatonin values start to be significantly different from daytime values at P8 (P8: *p* < 0.05; P15: *p* < 0.01) [189]. With kind permission.

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
