# Peer review of "Postnatal Development of the Circadian Rhythmicity of Human Pineal Melatonin Synthesis and Secretion (Systematic Review)"

_children, 2024, doi:10.3390/children11101197_

Round 1

Reviewer 1 Report

Comments and Suggestions for Authors

The subject is of interest; however, the paper requires revisions and improvements, see below

For simplicity I will start from technical issues.

Refence citation and video referral in the abstract are now allowed per MDPI journals specifications. I also believe that the number of allowed words has exceeded its limit.

The author focus on pineal gland may lead readers to wrong conclusions that this is the only site producing melatonin in the body.

Melatonin  is produced in extra-pineal sites including brain and importantly several extracranial sites. Examples are the GI, which can produce locally monumental amounts of melatonin or skin placed at the interface with the environment (Cell Mol Life Sci 74(21), 3913-3925, 2017).  This has to be mentioned and taken into con sideration.

Furthermore, melatonin and its metabolites are produced by diverse organisms (Biol Rev Camb Philos Soc2024 Apr 30. doi: 10.1111/brv.13091), which has to be mentioned since it is also produced by plants and some bacteria. 

The discussion on melatonin levels in the milk is important but it is also present together with serotonin is several crucial nutritional products such as honey (ACS Food Science & Technology 2021 1 (7), 1228-1235; DOI:10.1021/acsfoodscitech.1c00119; Melatonin Research. 5, 3 (Sep. 2022), 374-380. DOI:https://doi.org/https://doi.org/10.32794). This should be mentioned and discussed for obvious reasons in pediatric medicine.

The half-life of melatonin is very short. It is rapidly metabolized. Notably in the liver. In addition there is rapid metabolism of melatonin to several metabolites in the skin 

Comments on the Quality of English Language

Minor proof-reading is required that include corrections of of formatting and typographical errors

Reviewer 2 Report

Comments and Suggestions for Authors

The authors presented the review about postnatal development of the circadian rhythmicity of human pineal melatonin synthesis and secretion. The article is well written and of obvious interest to readers. I have some main comments:

1) Conclusion section needs to be added. Also formulate the main key points that follow from the review and are characterised by scientific novelty.

2) It would be good to add a sub-section in Discussion with key points about remaining unresolved issues, knowledge gaps.

Minor remarks:

1) Figures should be of better quality and contrast.

2) References in the Abstract are inappropriate.

Reviewer 3 Report

Comments and Suggestions for Authors

In this review on the role of melatonin in breast milk for the health and proper development of infants the author analyzed available literature on this topic, structured the contents for clinical relevance and draws conclusions of potential benefit for future early child care.

This is a sound analysis of the matter in which, however, the author understandably also advertises his patent application for melatonin-adjusted human or bovine "nightmilk". I am fine with that and, as it should be the author states this correctly in the manuscript.

I do not want to get too much into detail, but how exactly melatonin executes its actions is clear (via G-Protein coupled receptors) but there is also still the mention that melatonin acts as an antioxidant, which data mostly do not support (see here: Boutin & Jockers, Melatonin Controversies, an update, 2020 10.1111/jpi.12702).

Round 2

Reviewer 1 Report

Comments and Suggestions for Authors

The author was responsive to the critique and adequately replied to the points indicated.

Reviewer 2 Report

Comments and Suggestions for Authors

I approve the updated version of the article.